# Mifepristone (RU-486^®^) as a Schedule IV Controlled Drug—Implications for a Misleading Drug Policy on Women’s Health Care

**DOI:** 10.3390/ijerph19148363

**Published:** 2022-07-08

**Authors:** Yi-Ping Hsieh, Yun-Ju Wang, Ling-Yi Feng, Li-Tzy Wu, Jih-Heng Li

**Affiliations:** 1Department of Social Work, University of North Dakota, Grand Forks, ND 58202, USA; yiping66@gmail.com; 2College of Law, National Chung-Cheng University, Chia-Yi 62102, Taiwan; lawyjw@ccu.edu.tw; 3Doctoral/Master Degree Program in Toxicology, Kaohsiung Medical University, Kaohsiung 80708, Taiwan; joanna@kmu.edu.tw; 4Department of Psychiatry and Behavioral Sciences and Department of Medicine, School of Medicine, Duke University, Durham, NC 27710, USA; litzy.wu@duke.edu

**Keywords:** mifepristone (RU-486^®^), drug scheduling, abortion, stigma, human rights, drug policy

## Abstract

Background: Mifepristone (RU-486) has been approved for abortion in Taiwan since 2000. Mifepristone was the first non-addictive medicine to be classified as a schedule IV controlled drug. As a case of the “misuse” of “misuse of drugs laws,” the policy and consequences of mifepristone-assisted abortion for pregnant women could be compared with those of illicit drug use for drug addicts. Methods: The rule-making process of mifepristone regulation was analyzed from various aspects of legitimacy, social stigma, women’s human rights, and access to health care. Results and Discussion: The restriction policy on mifepristone regulation in Taiwan has raised concerns over the legitimacy of listing a non-addictive substance as a controlled drug, which may produce stigma and negatively affect women’s reproductive and privacy rights. Such a restriction policy and social stigma may lead to the unwillingness of pregnant women to utilize safe abortion services. Under the threat of the COVID-19 pandemic, the US FDA’s action on mifepristone prescription and dispensing reminds us it is time to consider a change of policy. Conclusions: Listing mifepristone as a controlled drug could impede the acceptability and accessibility of safe mifepristone use and violates women’s right to health care.

## 1. Introduction

Abortion is a complicated issue because it is often intertwined among health, law, and ethics. On average, 73.3 million induced (safe and unsafe) abortions occur worldwide each year, and 61% of all unintended pregnancies end in an induced abortion [1] Among induced abortions, more than half of all estimated unsafe abortions globally were found in Asia, especially in the south and central regions of Asia [2]. Mifepristone (RU-486^®^) was first marketed in France in 1990 [3]. With an efficacy of ca. 95–98%, oral intake of mifepristone alone or in combination with misoprostol can successfully interrupt early pregnancy up to 63 days (9 weeks) [3,4,5]. This medication-assisted abortion provides a safe and convenient alternative to vacuum aspiration or dilatation and curettage (D and C). Although mifepristone is considered a relatively safe and effective method for non-surgical abortion and a success from the viewpoints of medical achievements and public health to protect women’s health, there has been a debate over its use concerning the issue of “proper” abortion [6,7].

In Western countries, the main arguments of abortion are the right of pregnant women to have reproductive autonomy (pro-choice) and the right of the fetus to live (pro-life) [8,9,10]. While there has been a trend toward the liberalization of abortion laws [11,12,13,14,15], the debate over the moral and legal status of induced abortion still sides with either the pro-choice movement (i.e., emphasizing the right of women to decide whether to terminate a pregnancy) or the pro-life movement (i.e., emphasizing the right of the embryo or fetus to gestate to term and be born). Recently, such a standoff seems to have eased in the event of the COVID-19 pandemic. Due to the consideration of not putting patients and healthcare personnel at increased risk for COVID-19 by a clinic visit solely for dispensing the medication and a related petition from the American College of Obstetricians and Gynecologists (ACOG), the US Food and Drug Administration (FDA) has issued a notice to halt the enforcement of in-person dispensing of mifepristone (RU-486^®^) and allow dispensing through the mail either by or under the supervision of a certified prescriber, or through a mail-order pharmacy when the dispensing is done under the supervision of a certified prescriber [16]. Research data related to the in-person dispensing requirement in the Mifepristone Risk Evaluation and Mitigation Strategies (REMS) program during the COVID-19 pandemic have revealed no significant increase in safety concerns as a result of modifying the in-person dispensing requirement during the pandemic [17].

However, the progress of human rights protection in Taiwan differs from that in Western countries. This may be due to the fact that Taiwan underwent a Martial Law period from 1949 to 1987. When the Genetic Health Law was first enacted in 1984, Taiwan was still under the reign of Martial Law. Therefore, the legislators’ initial and main concern was not meant to ensure women’s reproductive autonomy and women’s rights. Instead, it was enacted mainly to promote the population health quality in addition to enhancing the health of mothers/children and family happiness. The following revisions of the Genetic Health Law (1999, 2009) [18] provided the exempt conditions for legal abortions. Furthermore, though not a member of the UN, Taiwan signed the Convention on the Elimination of all Forms of Discrimination against Women (CEDAW) in 2007 and enacted the Enforcement Act of CEDAW in 2012. The government has drafted the Reproductive Health Law to ensure women’s reproductive autonomy.

In contrast to the US FDA’s timely response to the petitions of the ACOG, the restriction policy on mifepristone regulation in Taiwan has raised concerns over the legitimacy of listing a non-addictive substance as a Schedule IV controlled drug, as it may affect women’s reproductive and privacy rights. In Taiwan, abortion remains a criminal offense for both abortionists and pregnant women unless it is performed under the conditions exempt from the Genetic Health Act (Legislation, 2009) [18]. These exemptions include the following: (1) a conception is due to incest or rape and (2) there is a diagnosis by a physician that the pregnant woman or her spouse has psychiatric disorders/hereditary defect(s), conception or delivery will result in a threat to the life of the pregnant woman, teratogenic formation is detected in the fetus, or conception or delivery will affect the psychological health or family life of the pregnant woman. Consequently, physicians in Taiwan play a crucial role in determining the legitimacy of abortion; however, they may fail to promote and protect women’s right to sexual and reproductive health due to the Genetic Health Act. It is important to note that mifepristone was developed to provide a non-invasive alternative for abortion for pregnant women, which may decrease the potential medical issues/harm and costs from surgical procedures performed by a physician. As a result of losing the option of using legal mifepristone for abortion in Taiwan, some pregnant women may seek illicit sources of mifepristone for abortion to protect their privacy.

It has been more than twenty years since mifepristone was listed as a Schedule IV controlled drug in Taiwan in 2000. According to the definition of the Controlled Drugs Act, the premise of a substance to be scheduled as a controlled drug is its dependence (i.e., addiction) potential. Mifepristone has not been known to have a potential for addiction. Taiwan’s tight control over mifepristone without addiction liability is unprecedented and may be unwarranted, which has produced an ongoing debate over the appropriateness of this policy on mifepristone-assisted abortion and its implications on women’s health and human rights [19,20,21,22,23,24]. The restriction policy of mifepristone used in Taiwan seems to be similar to that of its illicit drug policy. It is thus of interest to compare the policy and consequences of mifepristone-assisted abortion with those of illicit drug use.

## 2. Methods

Mifepristone was the first non-addictive medicine to be classified as a Schedule IV controlled drug in Taiwan. This article examines the following question: What connotations of social policy and human rights insights could be affected by the current policy of mifepristone-assisted abortion for pregnant women in Taiwan when compared with those of illicit drug use for persons with drug addiction? With the goal of developing appropriate social policies and effective implementation for women with unwanted pregnancies, we analyzed and examined the current policy to understand and improve policy contexts and consequences that could guide the decision of policy makers to reform the policies and ensure women’s rights and wellbeing in Taiwan. This article draws on a critical policy analysis (CPA) framework [25] to understand the complex connections between the current mifepristone-assisted abortion policy in Taiwan and the relations of dominance and subordination in the larger society in Taiwan. The legal status of mifepristone in Taiwan makes other types of studies difficult, if not impossible. CPA focuses on the policymaking process to explore all relevant aspects of the policy-making process, assumptions influencing policy implementation, and the sociopolitical and historical contexts in which policy are created [26]. Specifically, the CPA framework adopts the critical theory to examine how the policy discourse reflects the power dynamic in societies [27] for women with unwanted pregnancies. The CPA framework also draws on an institutional theory to examine whether administrative procedures are embedded in social and political environments [28]. This study utilizes a document analysis in qualitative research [29,30], including various acts and administrative decrees, UN conventions, and policies, to explore our research questions. The rule-making processes of mifepristone registration and scheduling were reviewed and analyzed. 

## 3. Results and Discussion

Table 1 illustrates the timeline of the associated legislations, administrative procedures, and policy discourse in Taiwan, which reflects the power dynamic in societies regarding women with unwanted pregnancies. 

This critical policy analysis examines how legislation portrays women with unwanted pregnancies, whether this policy possibly invades women’s reproductive health and privacy rights, and whether the policy-related social stigma and patriarchy influence the consequences. The policy and status of mifepristone use in Taiwan for unwanted pregnancies are compared with those of illicit drug use from various aspects of legitimacy, women’s human rights, social stigma, and access to health care.

### 3.1. Mifepristone as the First Non-Addictive Controlled Drug in Taiwan: A Case of the “Misuse” of “Misuse of Drugs” Laws

Before mifepristone was officially registered in Taiwan, the Medical Association of Gynecologists and Obstetricians (MAGO), stressing the danger of mifepristone-associated side effects, had proposed to the Department of Health (DOH) that the drug should be tightly controlled as a medical narcotic, which should be distributed and monitored by the National Narcotics Bureau [31,32]. The request was soon turned down because mifepristone is not a narcotic.

To comply with the three UN conventions on narcotics and psychotropic agents (United Nations, 1961; 1971; 1988) [33,34,35], Taiwan has enacted two anti-drug related laws: the Narcotics Hazard Prevention Act and the Controlled Drugs Act for the control of illegal and legal addictive substances in 1998 and 1999, respectively [36,37]. These two new laws, complementary to each other, follow the same drug scheduling system of the UN conventions that classifies addictive drugs into four schedules [33,34,35].

According to the Controlled Drugs Act, a controlled drug must possess three concurrent characteristics, i.e., addictive (or dependent) liability, abuse liability, and hazard liability to society [36,37]. Moreover, all controlled drugs, including narcotics, psychotropic agents, and other drugs where reinforced control is deemed necessary, are further divided into four schedules. Some representative items in these four schedules include: (1) Schedule I—opium, heroin, and cocaine; (2) Schedule II—amphetamines, cannabis, ecstasy, fentanyl, LSD, and pethidine; (3) Schedule III—flunitrazepam, ketamine, methylphenidate, and secobarbital; and (4) Schedule IV—most benzodiazepines, zolpidem, and zopiclone. The Controlled Drugs Act has authorized the National Bureau of Controlled Drugs (NBCD), which was later merged into the Taiwan Food and Drug Administration on 1 January 2010, to monitor the drug distribution and flow and to inspect medical records and the prescribing behaviors of physicians.

After the Act was promulgated, the MAGO lobbied some legislators to again petition the DOH to list mifepristone as a controlled drug [31]. The MAGO argued that mifepristone use could never be efficiently monitored if it was under a loose regulation system of prescription drugs [32]. Regardless of women’s rights groups’ opposition to the scheduling of mifepristone [38], this issue was brought up for discussion by the Committee for Evaluation of Controlled Drugs (CECD), which was composed of governmental representatives and academic experts for drug evaluation and scheduling. In the first two consecutive CECD meetings, the proposal to classify mifepristone as a controlled drug was denied because its non-addictive property did not meet the basic criteria of a controlled drug. However, mifepristone was classified as a schedule IV controlled drug at the third sequential meeting of the CECD [39]. Thus, although mifepristone did not fit the definition of a controlled drug by law, it became the first and the only non-addictive drug on the list of controlled drugs in Taiwan on 23 March 2001 [40].

When the Act for Prevention and Control of (Illicit) Drug Hazard and the (Licit) Controlled Drugs Act were first promulgated, all items in these four schedules were identical in both Acts. Thus, any changes of the scheduled items in one Act were simultaneously synchronized in the other Act. This consensus between the Ministry of Justice and the Department of Health had been maintained until the item of mifepristone was removed by the Ministry of Justice from the illicit drug schedule list on 9 January 2004 [41]. The Ministry of Justice also suggested that the DOH re-evaluate the suitability of mifepristone as a controlled drug, but the DOH maintained the decision.

In a commentary to a newspaper, the Taiwan Women’s Alliance, which was one of the major women’s rights organizations in Taiwan, criticized the statement of the “danger of mifepristone” as misleading [22]. In this article, the relative danger to health was compared between mifepristone (RU-486^®^) and sildenafil citrate (Viagra^®^). For mifepristone, there were only a few reported cases of non-specific fatal sepsis that could be related to women undergoing medical abortion in the US [42,43,44,45]. By contrast, sildenafil was linked to a total of 522 reported deaths in the US during a period of 13 months of availability [46]. The Taiwan Women’s Alliance challenged the perception that mifepristone was dangerous to women’s health, stating that Viagra^®^ would be more qualified to be classified as a controlled drug than mifepristone. Although the comparison of the health risks of RU-486^®^ and Viagra^®^ seems arbitrary, this comparison reveals the observation of political pressures or populism underlying the support for listing mifepristone as a Schedule IV drug in Taiwan. It is worth noting that the safety of mifepristone has been confirmed clinically [5,45,47,48]. Thus, the classification of mifepristone as a controlled drug was related to political decisions and/or populism rather than scientific evidence and professional judgment, which has important implications for women’s health and human rights issues [49]. This is especially critical for adolescent women who underwent abortions and experienced tremendous social and moral pressure from the traditional Taiwanese social-cultural environment [50].

By comparison, drug scheduling in Taiwan has been executed with extreme caution because the illicit use of Schedule I and II drugs is a criminal offense according to the Act for Prevention and Control of Illicit Drug Hazard [51]. For example, ketamine (a surgical anesthetic drug) has been used recreationally as a club drug, and is now on the list of Schedule III drugs. However, due to the increased use of ketamine for nonmedical reasons among illicit drug users, the Legislative Yuan (Congress) has emphasized the need to reschedule it to a Schedule I or II drug [52]. If ketamine was rescheduled, it would impact non-medical ketamine users who would be subject to urine screening, coercive treatment, and/or incarceration. Therefore, classifying mifepristone as a controlled drug not only negatively affects women’s access to mifepristone and reproductive autonomy, but also may create stigma for using mifepristone. The evaluation system for drug scheduling in Taiwan should be further scrutinized for safety concerns, because any inappropriate scheduling of a drug, entrusted by the law to the CECD, may cause unwanted consequences (e.g., turning the behavior of a drug user from a minor civil violation to a criminal offense).

### 3.2. Restriction Policy and Regulation on Abortion and Drug Use: The Legacy of Martial Law and Issue of Human Rights

The first version of the Act for Eradication of (Illicit) Narcotics was enacted in 1955 (at the early stage of Martial Law) with an aim to prevent the then rival Communist China from using narcotics as a weapon to weaken Taiwan’s combat capability [53]. Therefore, all drug-related offenses, either from the side of supply or demand, were strictly regulated. For instance, people with any use of heroin or cannabis could be sentenced to jail for 3–7 and 1–3 years, respectively.

After Martial Law was lifted in 1987, the jurisprudence in Taiwan developed more freely and began to emphasize “human rights.” Based on the protection that the Taiwan Constitution grants to the people, the restriction of people’s fundamental rights in the laws and regulations was re-examined for its constitutionality. However, both the issues of abortion and illicit drug use have continued to be impacted by the abolished Martial Law. By comparison, the issue of abortion in Western countries has developed from sociological and policy perspectives into a legal matter, and has been resolved in the highest courts, such as the Supreme Court of the United States and the Federal Constitutional Court of Germany [42]. By contrast, there have been no justice reviews on the abortion issue in Taiwan.

Regarding the case of illicit drug use, the illicit use of schedule I and II drugs under the new Narcotic Hazard Prevention Act indicates that an illicit user can be sentenced to jail for 6 months to 5 years and less than 3 years, respectively. For illicit use of schedule III and IV drugs, users can be fined within the range of TWD 10,000 to 50,000 and are required to take a 4–8 h compulsory drug education. The punitive policy for illicit drug use in Taiwan inherited the old Act for Eradication of (Illicit) Narcotics, which is part of the legacy of Martial Law. Therefore, the punitive nature of the policy for illicit drug use reveals the inappropriate classification of mifepristone as a controlled drug in Taiwan. 

Another related argument appeared in the debate concerning the right to use medicinal drugs [54]. The right to use medicinal drugs is granted in the case of saving a life or reducing pain, whereas there is no such right granted for non-medical use of drugs, because nonmedical (illicit) drug use could cause harm to the rights of others in some circumstances that could cause public health issues. However, according to the Genetic Health Law, the legitimacy of mifepristone use is determined by the diagnosis of a physician. In other words, the medical need to use mifepristone by women is often neglected and their reproductive autonomy is not respected under the obsolete Genetic Health Law. Thus, this restriction policy on abortion medication under the improper regulations in Taiwan is contradictory to the missions of the current drug control policy regarding its purpose of decriminalization and harm reduction. Taken together, listing mifepristone as a schedule IV drug in Taiwan illustrates the current policy’s failure to take human rights into consideration.

### 3.3. Social Stigma: Abortion vs. Illicit Drug Use

Women with unwanted pregnancies may resort to unsafe abortion when they cannot access safe abortion such as medical abortion (e.g., tablets) [55]. In addition to restrictive laws and additional requirements from spouse authorization/permission, social stigma is one of the barriers to accessing safe abortion for women. Social stigma refers to social disapproval of personal traits and beliefs that are against or deviant from social and cultural norms [56]. Social stigma leads people to judge an individual with deviant traits and exclude him/her from social and personal interactions [57], social support, or other services.

Abortion stigma is defined as “a negative attribute ascribed to women who seek to terminate a pregnancy that marks them, internally or externally, as inferior to ideals of womanhood” (p.628) [58]. Women with undesired pregnancy in Taiwan face discrimination and social stigma, which includes community judgment of abortion [22,32]. In Chinese culture, a pregnant woman who seeks abortion is usually viewed as a failure who committed lustful and irresponsible deeds that are against the cultural norms of being a virtuous woman [31]. Concealing abortion actually reinforces the perpetuation of stigma in a vicious cycle [58]. Moreover, the abortion-related stigma in Taiwan is often associated with the stereotype of having a large amount of careless sex [31]. Such social stigma and stereotypes of moral judgment contributed to policy makers’ concerns that the availability of “convenient abortion services” (i.e., mifepristone-assisted abortion) could lead to further careless and unprotected sex, which would then result in a greater need for abortion. Based on this false assumption/perception and perceived stigma, pregnant women wanting an abortion are blamed, and classifying mifepristone as a controlled drug in Taiwan is considered to prevent the reciprocal cycle of careless sex and abortion for women. However, the punitive policy on mifepristone-assisted abortion could create more social stigma and trigger pregnant women to seek private but unsafe abortion services as a solution. In other words, to avoid social stigma or abortion stigma attached to women with undesired pregnancies, these women tend to seek abortion care clandestinely to protect their privacy, even while medically safer options are available [55,59].

On the other hand, illicit drug users are viewed by some people as unreliable and dangerous [60,61], and illicit drug users may perceive fears, rejection, or unfair treatment from others [62]. Therefore, illicit drug users tend to avoid people, which may further lead to alienation, isolation, segregation, or marginalization [60,63]. In addition, illicit drug users who seek treatment may also face social stigma. For example, patients in methadone or buprenorphine treatment programs have experienced stigmatizing forces, such as chronic unemployment and financial dependency, criminal activity, homelessness, and minority group status [64]. Such social stigma could lead to additional barriers to receiving proper and timely treatment, social services, and other related support [64,65]. When people continue experiencing stigma or being treated punitively, they are more likely to continue with illegal or substance-abusing behaviors [66,67,68]. The struggles of illicit drug users to return to normality are influenced by drug-related social stigma, keeping secrets, and active drug use.

Mifepristone is classified as a schedule IV controlled drug (even though it is a non-addictive medicine), and the stigmas associated with using a controlled drug (i.e., mifepristone) for abortion among women with undesired pregnancies may prevent them from obtaining safe and proper treatments and related services. Therefore, in comparison with the punitive drug policy on controlled drug use that deters illicit drug users from seeking addiction treatment, the restriction policy on mifepristone-assisted abortion may similarly prevent pregnant women from seeking legitimate treatment.

Similar to heroin use, abortion in Taiwan is basically judged as an illegal deed and criminal offense except for the exempt conditions stated in the Genetic Health Act, by which physicians are authorized to decide the exemptions [69]. Therefore, a pregnant woman is basically not entitled to abortion. The exemptions stated in the Genetic Health Act are merely for physicians to avoid the penalty from the Criminal Law, not for pregnant women’s best benefits and rights. Although women’s social and economic circumstances are taken into consideration in exemptions, pregnant women in Taiwan are not allowed to make the judgments on these personal circumstances, and they are also required to obtain permission from their spouse or parents for the abortion. In the dominant notions of patriarchy, having the ability to make a decision for abortion could mean women’s independence from their husband/partner, reproductive freedom, and more economic autonomy [70]. To promote patriarchy, men in Taiwan might rely on the judicial system to give them the power to condemn and control women’s sexual and reproductive independence. Furthermore, because mifepristone is a controlled drug, pregnant women’s medical records are subject to inspection by the health authorities. Therefore, the current drug policies invade pregnant women’s privacy rights and confidentiality, which could lead to negative consequences and additional barriers to women’s utilization of sexual health care services [71]. Given that the social stigma on abortion is associated with pregnant women’s unwillingness to be inspected via their private medical records [71], they may turn to seeking illicit mifepristone to protect their privacy and so-called “secret”. Thus, instead of protecting women’s health care, the punitive policies may become barriers for delivering safe abortion services.

### 3.4. Restriction Policy and Access to Health Care: Mifepristone-Assisted Abortion vs. Drug Use

Placement of mifepristone as a controlled drug authorizes the government agencies to monitor its use. According to the Controlled Drugs Act in Taiwan, all prescriptions and medical records of controlled drugs are subject to inspection by the central and local health authorities. Although civil servants have the obligation to keep any private information confidential during their official duties, they are also required to report a criminal offense as per article 241 of The Code of Criminal Procedure. After mifepristone was classified as a controlled drug, a nationwide inspection was conducted from April 2001 to December 2003 to monitor its use. The results indicated that 0.13% (2/1540) of inspected pharmacies were found to sell illicit mifepristone without a prescription, while 2.63% (11/418) of inspected hospitals/clinics were also using illicit mifepristone to perform abortions [72]. The illicit use of mifepristone in hospitals/clinics was not as expected. Since abortion is a criminal offense except for exemptions described in the Genetic Health Law, a pregnant woman seeking abortion may worry about a possible invasion of their privacy. Such a mentality could explain why some pregnant women chose to use illicit mifepristone from either pharmacies or medical clinics to protect their privacy [32]. Acquirement of illicit mifepristone from other sources such as the internet or black market would pose an even greater health risk for pregnant women because of the unknown quality and safety of illicit sources of mifepristone. These data show the risk to women’s health of using mifepristone from illicit sources for undesired pregnancies under the tight-control policy in Taiwan. Although the Legislative Yuan passed the Enforcement Act of UN Convention on the Elimination of all Forms of Discrimination against Women (CEDAW) in 2012, the Taiwan Food and Drug Administration reiterated the notion that mifepristone, being non-addictive, is a schedule IV controlled drug and should be taken in front of a gynecologist/obstetrician in 2017 (Table 1).

Placing mifepristone as a schedule IV drug may have deterred some pregnant women from obtaining access to treatment. Such a phenomenon is similar to the consequences of the illicit use of addictive drugs and drug-related treatment-seeking behaviors.

Mifepristone was developed to give women a safe and effective alternative to terminate unwanted pregnancy. However, the issue of abortion is often complicated with issues related to ethics, law, and public health. As shown in Table 1, the complexity of these issues in Taiwan has been enhanced by the Martial Law, the legacy of which has included the Genetic Health Act and the Narcotics Hazard Prevention Act (and its counterpart the Controlled Drugs Act). Therefore, the restriction policy of misplacing mifepristone as a schedule IV drug by the petition of MAGO and the decision of DOH has created debatable punishments and social stigma. These negative aspects are deemed to discourage women from seeking safe and effective mifepristone-assisted treatment for unwanted pregnancy.

## 4. Conclusions

Drug scheduling is an essential tool adopted in the UN conventions for addictive drug control. In Taiwan, both the Act for Prevention and Control of Illicit Drug Hazard and the Controlled Drugs Act follow this basic principle of scheduling [73,74]. It is clear that mifepristone is not an addictive substance and should not be placed on the list of the three UN conventions on narcotics and psychotropic agents [33,34,35]. Therefore, the classification of mifepristone as a schedule IV drug is a violation of both laws, which are based on the three UN Conventions. As a result, the Ministry of Justice removed the item of mifepristone from the drug schedules of the Act for Prevention and Control of Illicit Drug Hazard on 9 January 2004 [41]. However, mifepristone has remained a schedule IV controlled drug in the Controlled Drugs Act mainly because of the Taiwanese MAGO’s concern over the danger of mifepristone misuse by pregnant women and the inability to efficiently monitor the use of mifepristone under a regulation system of prescription drugs [32]. The latter has influenced the policy-making of the DOH (now Ministry of Health and Social Welfare) in Taiwan. This restriction policy on mifepristone regulation reflects the sociopolitical and power dynamics in Taiwan, produces stigma, and negatively affects women’s reproductive and privacy rights. Such a restriction policy and social stigma may lead to the unwillingness of pregnant women to utilize safe abortion services. By contrast, in consideration of not putting patients and healthcare personnel at increased risk for COVID-19 because of making a clinic visit solely for dispensing the medication, the American ACOG has petitioned the US FDA to halt the requirement for in-person dispensing of mifepristone (RU-486^®^) and to allow dispensing through the mail either by or under the supervision of a certified prescriber, or through a mail-order pharmacy when the dispensing is done under the supervision of a certified prescriber [16]. In addition, studies during the past 20 years of mifepristone’s use in the US have demonstrated its effectiveness and safety [17]. The action taken by the US FDA on mifepristone prescription and dispensing during the COVID-19 pandemic has reminded us that it is time to consider a change in policy.

In summary, this paper shows that Taiwan’s restriction policy on mifepristone-assisted abortion via listing mifepristone as a controlled drug not only impedes the acceptability and accessibility of safe mifepristone use, but also violates women’s right to health. Such consequences can also be observed in the situation of the illicit use of addictive drugs and drug-related treatment-seeking behaviors when a punitive policy is implemented. The decision-making process of mifepristone registration and scheduling in Taiwan is unique in several aspects. First, classifying a non-addictive mifepristone as a schedule IV drug “misuses” the controlled drug policy/laws meant to protect public health. In conjunction with other studies on rational drug scheduling [75,76], it also implies that the current evaluation system for drug scheduling, both nationally and internationally, may need a comprehensive and objective remodeling or reshuffle to improve the clarity and safety for public health [74]. Second, the current restriction policy on mifepristone-assisted abortion prevents some pregnant women from receiving legitimate health care, which is similar to the punitive policy on illicit drug use that deters some illicit drug users from seeking timely addiction treatment. Third, the social stigma of mifepristone-assisted abortion attached to women with undesired pregnancies is similar to that of illicit drug use when mifepristone is a schedule IV drug. 

In “The Wind and the Sun,” one of Aesop’s fables [77], it was the warmth of the sun, not the chilliness of the wind, that won the traveler’s compliance. For pregnant women who have no other choice but to abort, embracing them with kindness and empathy may work better than isolating them with chilly punishment.

## Figures and Tables

**Table 1 ijerph-19-08363-t001:** Timeline of the associated legislation/regulation documents in Taiwan.

Year	Authority/Author(s)	Title	Key Points/Descriptions
1929	Legislative Yuan (Congress)	Controlled Narcotic Drugs Act	Availability of narcotic drugs for medical and scientific purpose Responsibilities for adequate use of narcotic drugs Criminal penalties
1935	Legislative Yuan	Offenses relating to Opium in Criminal Code	Offense of opium use, possession, supply, manufacture, and transport Introduction of criminal penalties
1949	Legislative Yuan	Temporary Act for Eradication of (Illicit) Narcotics (until 1952)	Offense of opium poppy cultivation Offense of opium use, possession, supply, manufacture, and transport Introduction of death penalty
1955	Legislative Yuan	Act for Eradication of (Illicit) Narcotics in the period of suppressing communist rebellion (special law)	Offense of narcotic drugs and synthetic congeners’ use, possession, supply, manufacture, transport, and opium poppy cultivation Introduction of death penalty Introduction of criminal penalties
1961	United Nations	Single Convention on Narcotic Drugs	International control on narcotic drugs
1971	United Nations	Convention on Psychotropic Substances	International control on psychotropic substances
1988	United Nations	Convention Against Illicit Traffic in Narcotic Drugs and Psychotropic Substances	International control on drug precursors
1998	Legislative Yuan	Narcotics Hazard Prevention Act	Scheduling of three categories of illegal narcotics by the Review Committee on Narcotics Introduction of differential penalties for offences of the first and second categories of narcotics Introduction of differential sanctions for offences of the third and fourth categories of narcotics
1999	Legislative Yuan	Controlled Drugs Act	Scheduling of four categories of Controlled Drugs (Legal Narcotics) by the Review Committee on Controlled Drugs Introduction of differential penalties for offences of the first and second categories of controlled drugs Introduction of differential sanctions for offences of the third and fourth categories of controlled drugs
2000	Department of Health, Executive Yuan, Taiwan	Minutes of the 5th meeting of the Review Committee on Controlled Drugs	Approval of mifepristone listed in the fourth category (Schedule IV) of controlled drugs
2000	Taiwan Women’s Alliance	Official letter to National Bureau of Controlled Drugs: On the management of RU-486 after its legalization	Opposition to the control of RU-486 as a controlled drug
2001	Decree of the Executive Yuan, Taiwan	No. 016828	Mifepristone listed in Schedule IV of controlled drugs
2003	Legislative Yuan	Narcotics Hazard Prevention Act (revised)	The fourth category of narcotics (for illegal purposes) added in the Narcotics Hazard Prevention Act to synchronize with the scheduling system of the Controlled Drugs Act Mifepristone listed in Category IV narcotics
2004	Decree of the Executive Yuan, Taiwan	Tai-Fa-Ji No. 0930001658	Mifepristone removed from Category IV narcotics
2009	Legislative Yuan	Genetic Health Act (revised)	Regulation of the conditions for legal abortions in order to enhance the health of mothers and children and advance family happiness
2012	Legislative Yuan	Enforcement Act of UN Convention on the Elimination of all Forms of Discrimination against Women (CEDAW)	Enforcement of CEDAW
2017	Taiwan Food and Drug Administration	Communication letter to all medical and pharmaceutical-related associations	TFDA reiterated that mifepristone, though not an addictive drug, remains on the list of schedule IV controlled drugs. It should be taken in front of a gynecologist/obstetrician for induced abortion.

## Data Availability

Not applicable.

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
