# Peer review of "Mifepristone (RU-486®) as a Schedule IV Controlled Drug—Implications for a Misleading Drug Policy on Women’s Health Care"

_ijerph, 2022, doi:10.3390/ijerph19148363_

Round 1
Reviewer 1 Report
The authors make a strong case for re-consideration and re-evaluation of the scheduling of Mifepristone from its current status as a Schedule IV drug because of its lack of potential for addiction. The issues are well presented and documented.
One consideration would be to include a figure that represents a timeline of the associated legislation, specifically affecting Taiwan, but also enabling a comparison with other countries in terms of how they have dealt with this issue
Reviewer 2 Report
The article represents a general analysis of the use of mifepristone in the women population and their right to have access to the drug. It seems an article highlights the right of women without providing a scientific insight on the matter. It would have been valuable, as a public health journal to ascertain if there is a relationship between doing a field analysis with women of reproductive age and correlating it with the prescription of the drug. In summary, the manuscript has to contain more than a general argument to be accessed for publication in a scientific journal
Reviewer 3 Report
Authors can present the methodology of the research in a comprehensive manner. It is quite hard to indicate. Social stigma and health issues of it can be described firther with a descriptivev manner and it should be also highlighted.
Reviewer 4 Report
I think the current manuscript is best presented as a commentary or a review article. The methodology of article inclusion, screening and critical appraisal is not clear.
Round 2
Reviewer 2 Report
The manuscript was improved partially although it was difficult to read since the authors should have cleaned the file and just highlight the new text. There are several grammatical errors in the inserted text to correct. The new table was appreciated, but the discussion requires work
Reviewer 4 Report
Thank you for the revised manuscript. I have no further comment.
Author Response
The authors thank the reviewer's encouragement.